# Zika virus infection differentially affects genome-wide transcription in neuronal cells and myeloid dendritic cells

Tamina Park[1,2], Myung-gyun Kang[1], Seung-hwa Baek[1,3], Chang Hoon Lee[4]*, Daeui Park[1,2,3]*

1 Department of Predictive Toxicology, Korea Institute of Toxicology, Daejeon, Republic of Korea, 2 Department of Human and Environmental Toxicology, University of Science and Technology, Daejeon, Republic of Korea, 3 Center for Convergent Research of Emerging Virus Infection, Korea Research Institute of Chemical Technology, Daejeon, Republic of Korea, 4 Bio and Drug Discovery Division, Center for Information-Based Drug Research, Korea Research Institute of Chemical Technology, Daejeon, Republic of Korea

* lee2014@krict.re.kr (CHL); daeui.park@kitox.re.kr (DP)

**Data Availability Statement:** The RNA-Seq data used in this study were SRA accession numbers SRP090990 (https://www.ncbi.nlm.nih.gov/sra?term=SRP090990) and SRP113558 (https://www.

## Abstract

Zika virus (ZIKV) is an emerging mosquito-borne flavivirus that has attracted global attention and international awareness. ZIKV infection exhibits mild symptoms including fever and pains; however, ZIKV has recently been shown to be related to increased birth defects, including microcephaly, in infants. In addition, ZIKV is related to the onset of neurological disorders, such as a type of paralysis similar to Guillain-Barré syndrome. However, the mechanisms through which ZIKV affect neuronal cells and myeloid dendritic cells and how ZIKV avoids host immunity are unclear. Accordingly, in this study, we analyzed RNA sequencing data from ZIKV-infected neuronal cells and myeloid dendritic cells by comparative network analyses using protein-protein interaction information. Comparative network analysis revealed major genes showing differential changes in the peripheral neurons, neural crest cells, and myeloid dendritic cells after ZIKV infection. The genes were related to DNA repair systems and prolactin signaling as well as the interferon signaling, neuroinflammation, and cell cycle pathways. These pathways were interconnected by the interaction of proteins in the pathway and significantly regulated by ZIKV infection in neuronal cells and myeloid dendritic cells. Our analysis showed that neuronal cell damage occurred through up-regulation of neuroinflammation and down-regulation of the DNA repair system, but not in myeloid dendritic cells. Interestingly, immune escape by ZIKV infection could be caused by downregulation of prolactin signaling including *IRS2*, *PIK3C3*, *JAK3*, *STAT3*, and *IRF1* as well as mitochondria dysfunction and oxidative phosphorylation in myeloid dendritic cells. These findings provide insight into the mechanisms of ZIKV infection in the host and the association of ZIKV with neurological and immunological symptoms, which may facilitate the development of therapeutic agents and vaccines.

ncbi.nlm.nih.gov/sra?term=SRP113558). In SRP090990, 13 SRA files were used for human pluripotent stem cell-derived hPNs and hNCCs infected with mock or Asian-lineage ZIKV (PRVABC59 isolated from Puerto Rico in 2015). In SRP113558, we used 23 SRA files with RNA-Seq results using purified mDCs isolated from three patients with naturally-acquired acute ZIKV infection and from 20 healthy individuals.

**Funding:** This work was supported by National Research Council of Science & Technology grant by the Korea government (CRC-16-01-KRICT) to DP.

**Competing interests:** The authors have declared that no competing interests exist.

## Introduction

Zika virus (ZIKV) disease is an epidemic caused by ZIKV, an icosahedral enveloped arbovirus with a positive-sense, single-stranded RNA genome of approximately 11 kb. ZIKV belongs to the genus *Flavivirus* and family *flaviviridae*, which includes dengue virus, yellow fever virus, and West Nile virus [1]. ZIKV is transmitted to humans mainly through the bite of infected *Aedes aegypti* and *Aedes albopictus* (Asian tiger mosquito). Additionally, ZIKV-infected men can transmit the virus to their sexual partners [2]. ZIKV strains are grouped into two major lineages, African and Asian, as confirmed by phylogenetic analyses [3]. Recent studies showed that the Asian lineage of ZIKV is strongly associated with microcephaly and Guillain-Barré syndrome [4,5]. Additionally, the marked increase in neonates born with microcephaly in northeast Brazil was shown to be caused by intrauterine exposure to ZIKV [6].

Most studies of the pathogenesis of ZIKV have focused on the central nervous system using human pluripotent stem cell-based models, such as human neural crest cells (hNCCs) and human peripheral neurons (hPNs) [7]. Furthermore, current *in vivo* research of ZIKV-infected mouse brains demonstrated that ZIKV infects cells during different brain maturation stages to induce changes in cortical tissue organization (e.g., reduced cell numbers and cortical layer thickness) [8]. ZIKV has been shown to target cortical progenitor cells and cause microcephaly by inducing cell death [9]. Moreover, in a recent report, ZIKV replication was detected in some dendritic cells, and myeloid dendritic cells (mDCs) circulating in the peripheral blood were susceptible to ZIKV during pregnancy and in infants [10,11]. These studies strongly suggest that both neuronal cells and immune cell play critical roles in ZIKV infection.

However, the mechanisms through which ZIKV infection affect hPNs, hNCCs, and mDCs are unclear. Therefore, studies are needed to identify the different mechanisms of neuronal cells and mDCs in the pathogenesis of ZIKV.

Accordingly, in this study, we compared the changes in genomic-wide gene expression in ZIKV-infected hPNs, hNCCs, and mDCs using publicly available archive data. We investigated the various mechanisms of ZIKV infection including the mechanism of neuronal cell death and remarkably distinguishable immunological changes in neuronal cells and mDCs. Particularly, ZIKV induced down-regulation in the expression of DNA repair system-related genes in neuronal cells, but not in mDCs. Interestingly, ZIKV-infected mDCs showed downregulation of prolactin signaling, mitochondrial dysfunction, and oxidative phosphorylation, but not in peripheral neurons and neuronal crest cells. Based on previous reports [12–14], mitochondrial dysfunction and oxidative phosphorylation can lead to escape of the immune defense in mDCs. We also observed differential changes in gene expression patterns related to inflammation between neuronal cells and mDCs. Taken together, ZIKV infection causes distinguishable changes in gene expression on neuronal cells and mDCs in systemically differential manner for neuronal cell death and the acquisition of immune suppression and escape capacity. These results strongly suggest that mDCs are critical cells targeted by ZIKV for the immune escape mechanism of ZIKV in infected hosts.

## Materials and methods

### RNA-sequencing data for cells infected with PRVABC59, Asian-lineage ZIKV

RNA-sequencing (RNA-Seq) data from various cell types infected with ZIKV were retrieved from the publicly available Sequence Read Archive (SRA), the primary archive of raw high-throughput sequencing data from the National Institutes of Health (https://www.ncbi.nlm.nih.gov/sra). The RNA-Seq data used in this study were SRA accession numbers SRP090990 and

SRP113558 [7,11]. In SRP090990, 13 SRA files were used for human pluripotent stem cell-derived hPNs and hNCCs infected with mock or Asian-lineage ZIKV (PRVABC59 isolated from Puerto Rico in 2015). In SRP113558, we used 23 SRA files with RNA-Seq results using purified mDCs isolated from three patients with naturally-acquired acute ZIKV infection and from 20 healthy individuals. These patients were three female adults who traveled to Caribbean destinations including Puerto Rico during autumn/winter 2015/2016 [11]. These 36 SRA files were evaluated using Illumina sequencing instruments and were preferentially downloaded using the SRA toolkit (https://www.ncbi.nlm.nih.gov/sra/docs/toolkitsoft/).

## RNA-Seq analysis to identify differentially expressed genes

The downloaded SRA files were converted to FASTQ files with fastq-dump packaged in the SRA toolkit 2.6.2, and quality was checked using fastQC v0.11.5 (http://www.bioinformatics.babraham.ac.uk/projects/fastqc/). The sequencing reads in FASTQ files were aligned to the NCBI human genome sequence, Genome Reference Consortium Human Build 38 patch release 12 (GRCh38.p12) using Spliced Transcripts Alignment to a Reference 2.4.1c [15]. The NCBI human genome annotation general feature format 3 file was also used for mapping because the raw sequence reads downloaded from SRA used human-derived or induced cell lines for sampling. Next, we used featureCounts with the Binary Alignment/Map file from the alignment result. featureCounts is a useful program for counting reads, read summarization, and assigning fragments to the gene [16]. Finally, the differentially expressed genes (DEGs) were identified as those showing a $\log_2$(fold-change) of greater than 1 and false discovery rate of less than 0.05 using EdgeR in Bioconductor software (https://www.bioconductor.org/) for counting data with an over-dispersed *Poisson* model and an empirical *Bayes* procedure [17]. The overall procedures were based on a previously described approach [18].

## Heat maps of representative pathways containing DEGs

To determine the significant biological pathways associated with gene sets containing DEGs as a result of RNA-Seq, representative pathways were analyzed by performing Ingenuity Pathway Analysis (IPA, http://www.ingenuity.com). Enrichment analysis was performed to analyze the gene sets generated by RNA-Seq, and biological pathways were represented based on enrichment analysis with DEGs. IPA was used to search critical pathways. Next, the DEGs were subjected to Gene Ontology analysis to identify different changes in cellular compartments between neuronal cells and mDCs using Enrichr [19]. The statistical significance (*p* value) of each pathway was calculated using Fisher's exact test. Heat map analysis was performed for the highest 10 pathways among pathways which were filtered according to *p* values of less than 0.01. However, the significant value of cellular component was less than 0.05. To compare the similarities in significantly expressed pathways among three samples, we used Euclidean distances and complete linkage as the agglomeration method for hierarchical clustering. Heat map images were generated in R with the heatmap.2 function in the gplots library (https://cran.r-project.org/web/packages/gplots/).

## Construction of a network among pathways

To observe the relationships among biological pathways of DEGs in ZIKV-infected cells, we constructed a pathway network based on protein-protein interaction information from DEGs. Significant biological pathways were selected from heat maps, and the relationships between pathways were constructed from the interactions of DEGs contained in these pathways. The interactions of DEGs were established using the STRING database [20], which contains information on protein interactions in humans. The STRING database provides not only

information regarding protein interactions but also a combined score indicating the probability of the interaction. The threshold for significant interaction pairs was set to a combined score of greater than 900. In the network, the color density of each node represents the statistical significance (*p* value) of the pathway. The node size was determined by betweenness centrality, which indicated the influence of communication among pathways. Pathways with a high betweenness centrality are interesting because they are on the communication path and can control information flow. The pathways are important among various biological pathways and are candidate targets for drug discovery [21]. The pathway network and *p* value of each pathway were visualized with Cytoscape version 3.7.0 software, an open source platform (http://www.cytoscape.org/).

## Comparative subnetwork analysis

To identify significant DEGs between neuronal cells and mDCs, we generated a comparative subnetwork. The candidates for the subnetwork were selected as significant pathways with *p* values lower than 0.05 in common among pathways up- or downregulated in neuronal cells and mDCs (S1 Fig). To improve the reliability of the results, statistical *t*-tests for DEGs were performed using the read counts obtained from RNA-Seq. A total of 1804 genes with a *p* value less than 0.05 (S1 Table) were screened and used to network the genes involved in the final top 10 pathways. The 10 pathways were integrated to create two representative subnetworks using pathways showing opposite regulation in neuronal cells and mDCs, e.g., upregulation in neuronal cells and downregulation in mDCs or downregulation in neuronal cells and upregulation in mDCs. In two representative subnetworks, the interactions between DEGs were derived from resources with a combined score of greater than 900 in the STRING database. In the subnetwork, a node was represented as DEGs showing fold-changes in two neuronal cells and myeloid dendritic cells. Additionally, fold-changes in each gene are presented as bar charts for hPNs, hNCCs, and mDCs infected with ZIKV. Fold-changes were calculated using $\log_2$ (fold-change).

## Results

### DEG profiling in two human neuronal models and myeloid dendritic cells

Among ZIKV-related RNA-Seq datasets in the public archive, we merged three datasets for neuronal cells and myeloid dendritic cells, i.e., hNCCs, hNCC-derived hPNs (SRP090990), and mDCs (SRP113558). Detailed experimental information regarding the datasets is described in Fig 1A. We identified DEGs through RNA-Seq analysis to examine differences in the characteristics in ZIKV-infected neuronal cells and mDCs. The analysis identified significantly altered genes in ZIKV-infected cells compared to in mock-infected cells (Fig 1B). Based on the criteria of $\log_2$ (fold-change) greater than 1 and false discovery rate of less than 0.05, 1171 genes were found to be differentially expressed in ZIKV-infected hPNs (604 upregulated genes and 567 downregulated genes), 1672 genes were differentially expressed in ZIKV-infected hNCCs (887 upregulated genes and 785 downregulated genes), and 2601 genes were differentially expressed in ZIKV-infected mDCs (1239 upregulated genes and 1362 downregulated genes). Venn diagrams were generated to compare the number of DEGs in the three different cell types (Fig 1C). The neuronal cells and mDCs shared only 2.9% genes in upregulated DEGs and 1.4% genes in downregulated DEGs. In the comparison of two neuronal cells, hPNs shared 39.4% (238/604) of genes in upregulated DEGs and 25.6% (145/567) of genes in downregulated DEGs with hNCCs. Additionally, hNCCs shared 26.8% (238/887) genes among upregulated DEGs and 18.5% (145/785) genes among downregulated DEGs with hPNs. All DEGs are described in S2 Table.

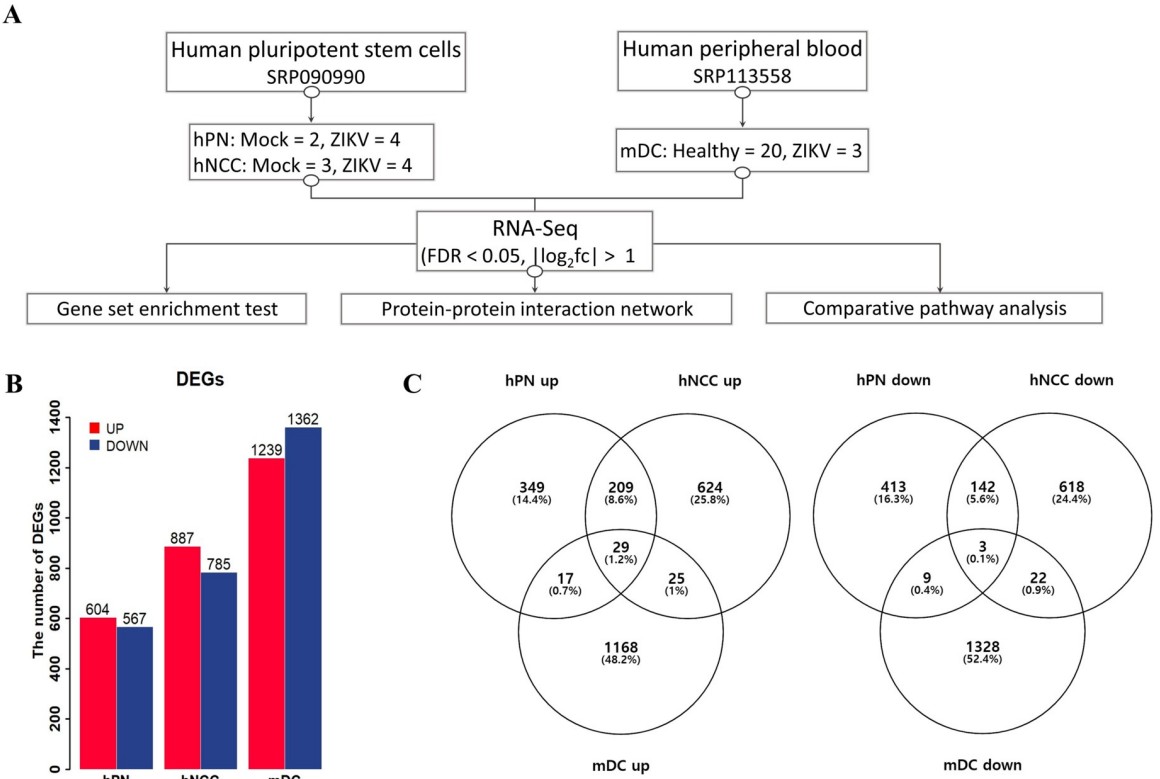

**Fig 1. Comparison of cell-specific gene expression between ZIKV-infected cells and uninfected cells.** (A) Simple schematic for the three integrated datasets, including three different cell types (hPNs, hNCCs, mDCs). (B) Histogram demonstrating the numbers of DEGs in hPNs, hNCCs, and mDCs. (C) Venn diagrams showing overlap in the number of up- or downregulated DEGs for the three cell types. hPNs: human peripheral neurons, hNCCs: human neural crest cells, mDCs: myeloid dendritic cells.

## Comparison of significant biological pathways between neuronal cells and myeloid dendritic cells

DEGs were grouped into biological pathways by performing gene set enrichment analyses. The results of RNA-Seq analysis showed that different mRNA profiles depended on the cell type after ZIKV infection. The representative terms for biological pathways were obtained from many journal articles, books, and HumanCyc, an alternative to KEGG pathway analysis using IPA. In gene set enrichment analyses, we identified 43, 49, and 181 significantly upregulated pathways in hPNs, NCCs, and mDCs, respectively, and 28, 11, and 44 significantly downregulated pathways in hPNs, hNCCs, and mDCs, respectively. After the top 10 significant pathways were selected under each condition, 25 upregulated pathways and 29 downregulated pathways were obtained by comparing neural cells and mDCs (Fig 2). Genes and *p* values for the pathways are shown in S3 Table.

Specifically, there were marked differences between neuronal cells and myeloid dendritic cells. Genes related to the "Neuroinflammation Signaling Pathway" were upregulated in hPNs and hNCCs compared to in uninfected cells, but were downregulated in mDCs (Fig 2A). Genes related to "Oxidative Phosphorylation" and "Mitochondrial Dysfunction" were upregulated only in mDCs (Fig 2A). Cell cycle pathways, such as "Role of CHK Proteins in Cell Cycle Checkpoint Control" and "Cell Cycle Control of Chromosomal Replication" were significantly downregulated in neuronal cells but were not downregulated in mDCs (Fig 2B). Additionally, DNA repair systems, including the "Nuclear Excision Repair (NER) pathway", "Mismatch

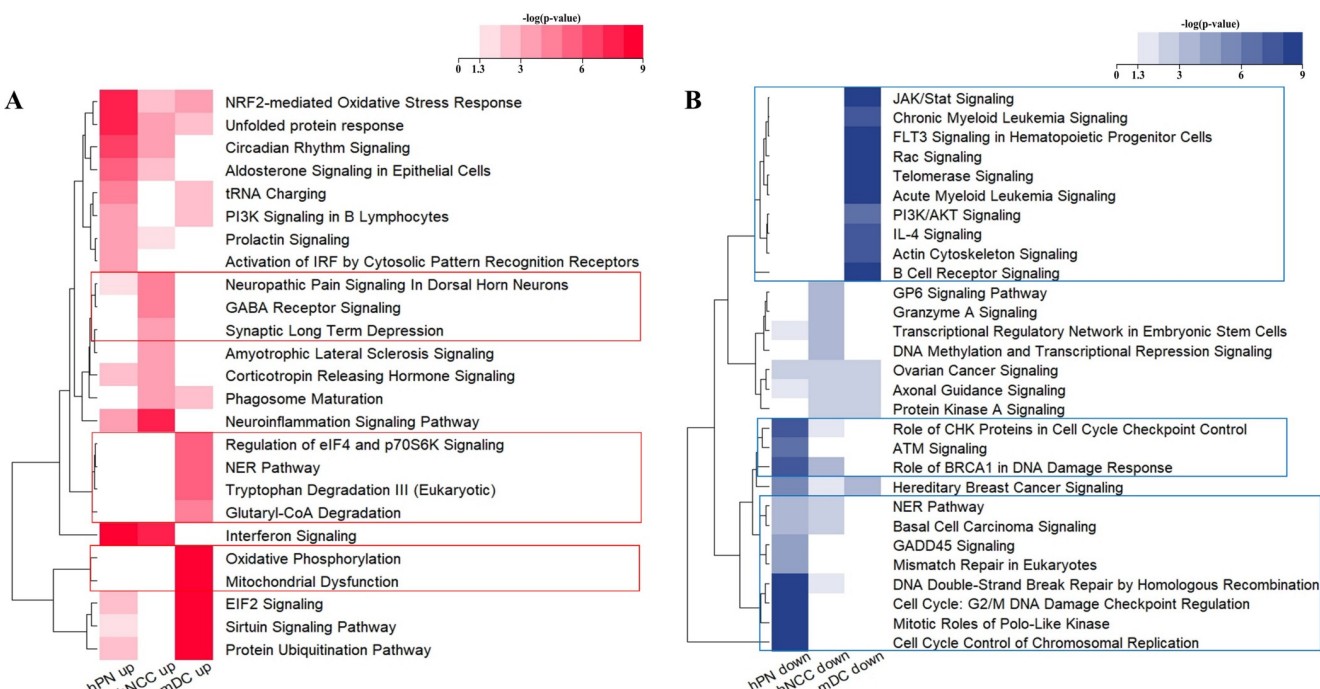

**Fig 2. Heat maps for comparative analysis of significant biological pathways in ZIKV-infected cells.** Heat maps were generated using Fisher's exact tests (*p* value) for pathways from gene set enrichment analysis. The values were converted to–$\log_{10}$ (*p* value), indicating the significance of the biological pathways containing DEGs. A value of 1.3 indicates a *p* value of 0.05. Additionally, the boxed line shows the specifically altered pathways in macrophages. (A) Heat map showing upregulated biological pathways. (B) Heat map showing significantly downregulated biological pathways. The heat maps were generated using R package.

Repair in Eukaryotes", "DNA Double-Strand Break Repair by Homologous Recombination", and "Cell cycle: $G_2$/M DNA Damage Checkpoint Regulation" were downregulated only in neuronal cells by ZIKV infection (Fig 2B).

Interestingly, the "Role of BRCA1 in DNA Damage Response", "ATM Signaling", and "GADD45 Signaling" were only downregulated in hPNs and hNCCs (Fig 2B). These pathways include upregulated *GADD45A* only in neuronal cells (hPNs: 1.62, hNCCs: 1.51 $\log_2$[fold-change], and mDC: not changed) and *GADD45* which function as stress sensors, resulting in cell cycle arrest, DNA damage responses, and apoptosis [22]. A previous study showed that ZIKV infection dysregulates human neural stem cell growth with upregulated *GADD45A* and other genes associated with cell cycle arrest [23].

## Interconnections of biological pathways by ZIKV infection

To understand the relationships among significantly altered pathways, we constructed a protein-protein interaction network grouped by biological pathways. The interactions of pathways were defined as the interaction of DEGs in the pathway. The top 10 pathways were selected based on *p* values in the up- or down-regulated pathways of each condition to normalize the number of pathways in three cell lines. The final network consisted of 53 pathways and 1181 interactions. As shown in Fig 3, the pathways with insignificant *p* values greater than 0.05 were removed. Relationships among pathways were emphasized by the thickness of the edges, which reflected the ratio of the number of DEG pairs interacting between the two pathways. The thickness of the edges indicates which pathways are closely and strongly interconnected with each other.

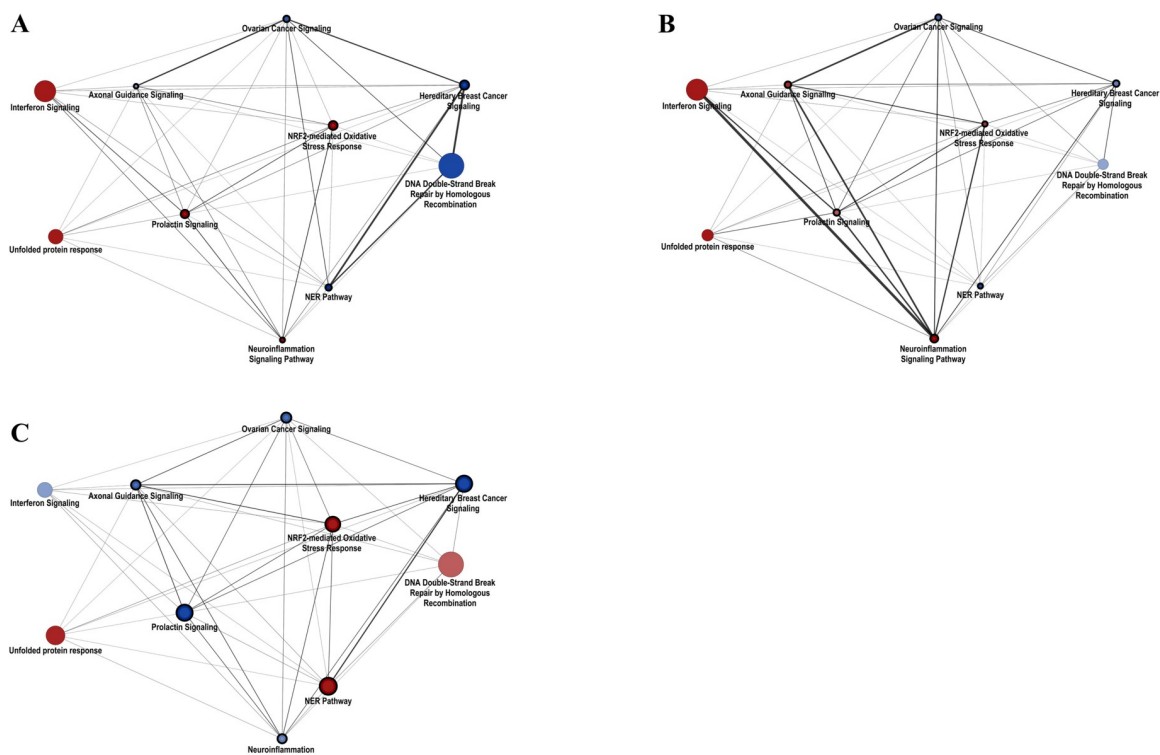

**Fig 3. Differential changes in pathway networks among neuronal cells and myeloid dendritic cells.** Protein-protein interactions were grouped by significantly expressed biological pathways in hPNs (A), hNCCs (B), and mDCs (C). The nodes represent biological pathways, and node size depends on the betweenness centrality score calculated by network analysis. Edges were generated based on interactions of DEGs contained in pathways. (A) The network with significantly expressed pathways (B) The network with significantly expressed pathways in hNCCs (C) The network with significantly expressed pathways in mDCs. Red: upregulated pathways; blue: downregulated pathways. The networks were generated using Cytoscape v3.7.0 software.

In hPNs (Fig 3A), the network showed that "Interferon Signaling", "tRNA charging", and "Circadian Rhythm Signaling" were upregulated with high betweenness centrality, and that downregulated genes involved in cell cycle and DNA repair system-related pathways strongly communicated with each other. "Interferon Signaling", "Neuroinflammation Signaling Pathway", "Phagosome Maturation", and "Neuropathic Pain Signaling in Dorsal Horn Neurons" were upregulated and strongly communicated with each other in hNCCs (Fig 3B). The networks showed that the downregulated cell cycle and DNA repair system and upregulated neuroinflammation responses have important roles in both hPNs and hNCCs. The results supported those of a previous study demonstrating that both hPNs and hNCCs were induced to cause massive cell death after ZIKV infection [7]. Additionally, in hPNs, the interaction between the cell cycle and DNA repair system showed a greater reduction than in hNCCs, whereas neuroinflammatory responses were higher than in hNCCs.

In the network of mDCs, most pathways with high betweenness centrality were unlike neuronal cells in which characteristic pathways are expressed (Fig 3C). Among the significantly changed pathways, DNA repair-related pathways such as "DNA Double-Strand Break Repair by Homologous Recombination" and "NER" were upregulated by ZIKV infection as defense mechanism against DNA damages. These defense mechanisms were observed in only mDCs, but not in neuronal cells. Additionally, "Oxidative Phosphorylation", "Mitochondrial Dysfunction", and "Sirtuin Signaling Pathway" were highly interconnected with each other for defense mechanism against ZIKV in mDC. The "NRF2-medidated Oxidative Stress Response" was

upregulated as a defense mechanism which serves to limit oxidative stress. Interestingly, "Prolactin Signaling" was downregulated in mDC by ZIKV infection, but not in neuronal cells. Prolactin in the immune system is known to stimulate the secretion of other cytokines and expression of cytokine receptors including IFN-gamma, IL-1, and IL-10 [24].

Also, commonly affected pathways after ZIKV infection in both neuronal cells and mDCs were cell cycle pathways ("Ovarian Cancer Signaling" and "Hereditary Breast Cancer Signaling"), immune response pathways ("Interferon Signaling", "Prolactin Signaling", and "Neuroinflammation Signaling Pathway"), and DNA repair system ("DNA Double-Strand Break Repair by Homologous Recombination" and "NER Pathway"), and a simplified network was shown between them (S1 Fig). In S1 Fig, cell cycle-related pathways were downregulated in both neuronal cells and mDCs. The major genes including pathways were *WNT2B*, *WNT5A*, *FZD1*, *FZD2*, *FZD3*, *TCF7L2*, and *RB1* which can affect cell growth and fate with the Wnt signaling pathway. The Wnt signaling pathway regulates brain development, cellular growth, tumorigenesis, and stem cell biology [24–27].

## Distinguishable differences from neuronal cells and mDCs

The most distinguishable differences between the two neuronal cell types and mDCs may be representative subnetworks. Immune response pathways such as "Interferon Signaling", "Prolactin Signaling", and "Neuroinflammation Signaling Pathway" were upregulated in neurons but downregulated in mDCs. "DNA Double-Strand Break Repair by Homologous Recombination" and "NER Pathway", which are downregulated in neurons but upregulated in mDCs, can be designated as DNA repair systems (Fig 4).

The two neuronal cell types commonly showed upregulation of immune-related genes, whereas mDCs showed downregulation of immune-related genes (Fig 4A). Specifically, in mDCs, but not in neuronal cells, we observed downregulation of genes related to cytokine-mediated signaling: *IRS2* (hPNs: 1.31, hNCCs: 1.16, mDCs: −0.35 log$_2$[fold-change]), *SOCS5* (hPNs: 0.77, hNCCs: 0.33, mDCs: −1.27), and *RELA* (hPNs: 0.37, hNCCs: 0.26, mDCs: −1.09);

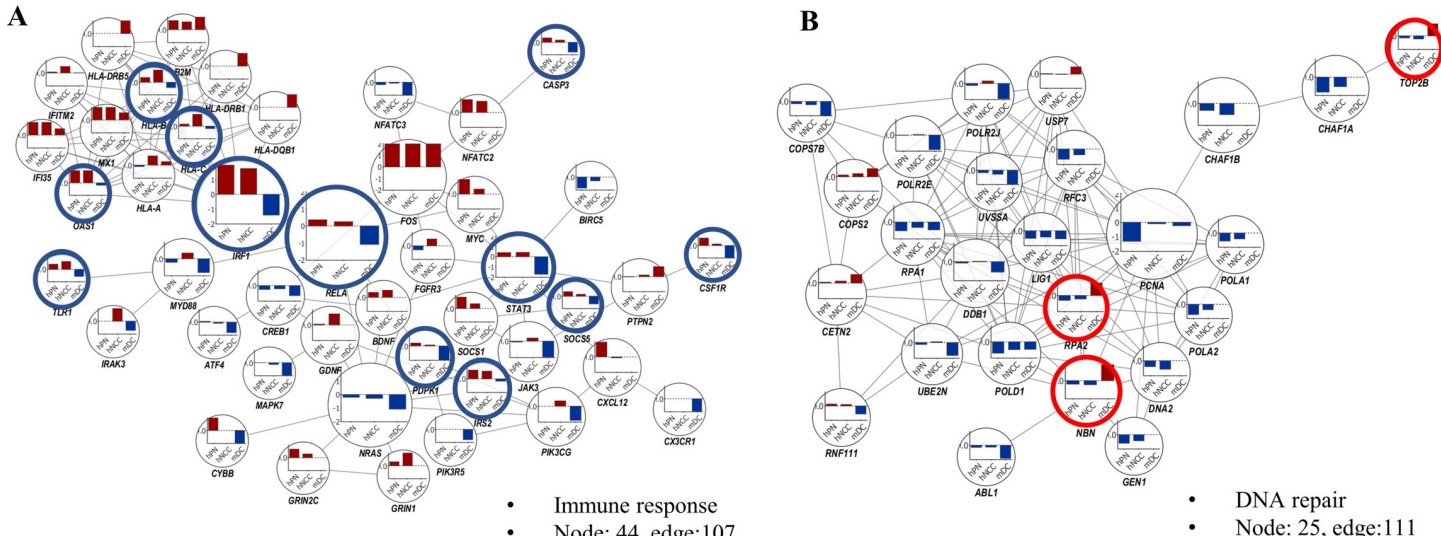

**Fig 4. Comparative subnetwork with DEGs and fold-changes in hPNs, hNCCs, and mDCs.** (A) The 107 protein interactions of 44 DEGs in cytokine-mediated signaling pathway. (B) The 111 protein interactions of 25 DEGs in DNA repair system. The bar charts in node show the fold-changes in hPNs (left bar), hNCCs (middle bar), and mDCs (right bar) infected with ZIKV. Fold-changes were calculated using log$_2$ (fold-change). Red circle: genes that are upregulated in mDCs as opposed to hPNs and hNCCs; blue circle: genes that are downregulated in mDCs as opposed to hPNs and hNCCs.

interferon-mediated signaling genes *OAS1* (hPNs: 4.28, hNCCs: 5.03, mDCs: −0.37), *IRF1* (hPNs: 2.56, hNCCs: 1.77, mDCs: −1.41), *STAT3* (hPNs: 0.41, hNCCs: 0.43, mDCs: −1.72); antigen processing and presentation genes *HLA-B* (hPNs: 0.79, hNCCs: 2.10, mDCs: −0.86); and *HLA-C* (hPNs: 0.36, hNCCs: 1.94, mDCs: −0.39), as shown in Fig 4B. Most genes upregulated in neuronal cells but downregulated in mDCs were associated with cytokine receptor or cytokine-mediated signaling.

Interestingly, "Prolactin Signaling" consisting of JAK/Stat signaling, PI3K signaling, and RAS cascade [25] were downregulated only in mDCs. PI3K signaling genes included *IRS2* (hPNs: 1.31, hNCCs: 1.16, mDCs: −0.35 $\log_2$ [fold-change]), *PIK3R5* (hPNs and hNCCs: not changed, mDCs: −1.61), *PIK3C3* (hPNs: 0.09, hNCCs: 0.45, mDCs: −1.20), *PIK3CG* (hPNs: not changed, hNCCs: 0.69, mDCs: −1.81), and *PDPK1* (hPNs: 0.43, hNCCs: 0.16, mDCs: −2.08). JAK/Stat signaling genes were *JAK3* (hPNs: −0.11, hNCCs: 0.40, mDCs: −1.95), *STAT3* (hPNs: 0.41, hNCCs: 0.43, mDCs: −1.72), and *IRF1* (hPNs: 2.56, hNCCs: 1.77, mDCs: −1.41). RAS cascade genes were *NRAS* (hPNs: −0.21, hNCCs: −0.29, mDCs: −1.09), *PRKCG* (hPNs: −1.09, hNCCs: 1.07, mDCs: not changed), *MYC* (hPNs: 2.52, hNCCs: 0.70, mDCs: not changed). However, two neuronal cell types commonly showed downregulation of genes related to the DNA repair system, including *TOP2B* (hPNs: −0.32, hNCCs: −0.46, mDCs: 1.80), *RPA2* (hPNs: −0.62, hNCCs: −0.44, mDCs: 1.57), and *NBN* (hPNs: −0.39, hNCCs: −0.46, mDCs: 1.78); in contrast, mDCs showed upregulation of these gene sets (Fig 4B). Prolactin signaling is related to stimulate the secretion of cytokines and expression of cytokine receptors in the immune system [34] which is consisted of PI3K signaling and JAK/Stat signaling (S2 Fig) Interestingly, the *IRS2*, *PIK3C3*, *JAK3*, *STAT3*, and *IRF1* which belong to prolactin signaling was significantly downregulated only in mDCs, not neuronal cells.

## Cell compartment changes between neuronal cells and mDCs

To better understand the effect of cellular compartments by ZIKV infection, we performed gene ontology analysis of DEGs using Enrichr. Cellular compartment analysis revealed 30 gene ontology terms related 681 DEGs (Fig 5). In neuronal cells, up-regulated genes were abundant in clathrin-sculpted vesicles (GO: 0061202, GO: 0061200), 'phagocytic vesicle membrane' (GO: 0030670), and 'recycling endosome membrane' (GO: 0055038), whereas this was not observed in mDCs.

ZIKV is introduced into host cells via clathrin-mediated endocytic mechanisms after its binds to cell surface receptors, after which it is transported to endosomes [26]. After entering the cell, viral RNA is translated to a larger polyprotein on the endoplasmic reticulum. After a few steps, the newly synthesized virus traffics from the endoplasmic reticulum to the surface through the Golgi to spread the infection to nearby cells [27]. During trafficking from the Golgi to the surface, the tubular trans Golgi network generates clathrin-coated vesicles for delivery to the plasma membrane in the secretory pathway [28]. Our result support that ZIKV can lead to activation of phagocytosis including phagosome, endoplasmic reticulum, and clathrin-mediate vehicle in neuronal cells.

## Discussion

In this study, we investigated the expression of ZIKV-induced genes in neurons, as the primary target cells of ZIKV infection and main cause of neuronal cell death. Two subspecies of neuron cells were evaluated to identify the cellular networks involved in ZIKV infection. In addition, ZIKV-infected mDCs analyzed by RNA-Seq from other sources were used to identify the detailed mechanisms of dysfunction in the immune response by ZIKV infection and for comparison with neuronal cells. In neuronal cells, inflammation-

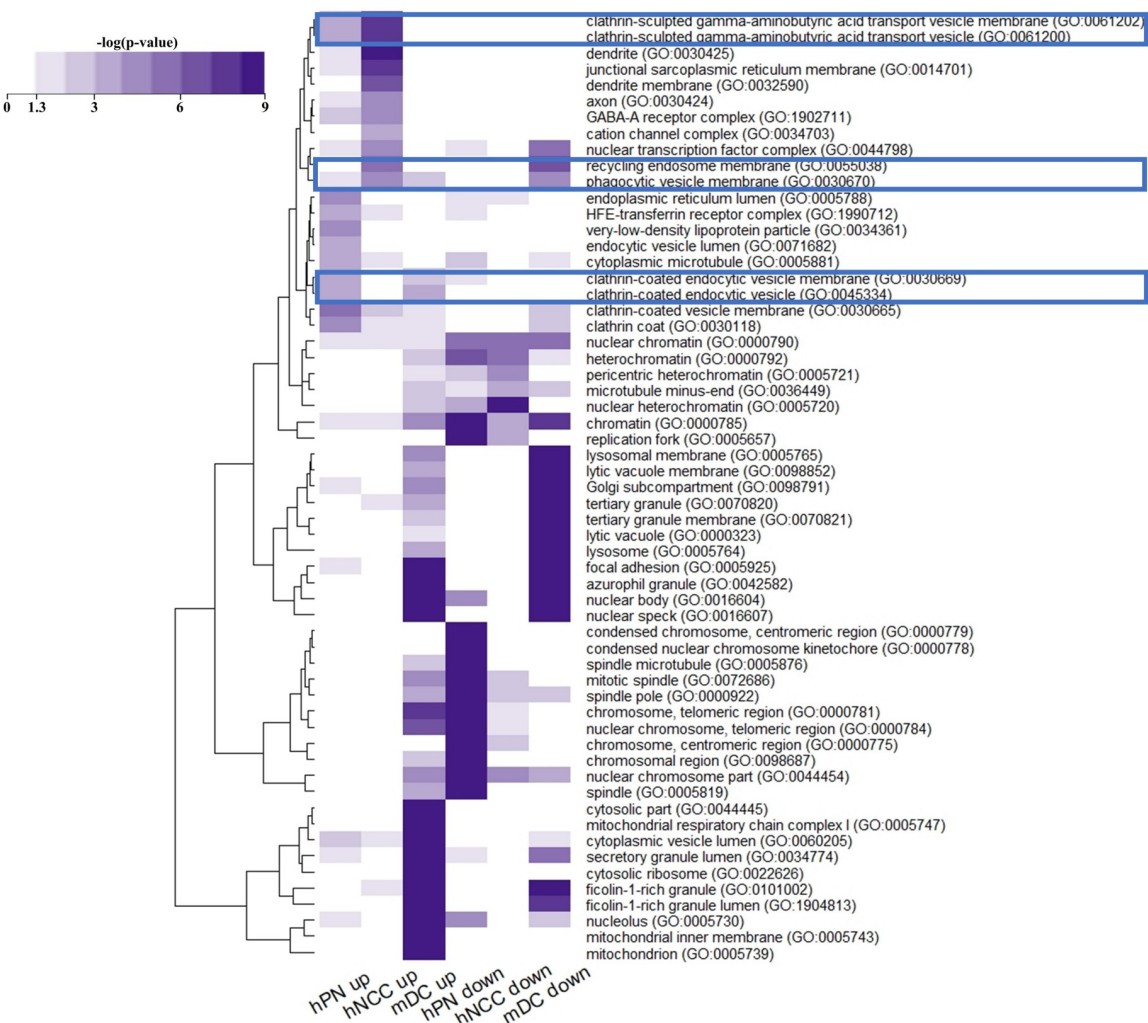

**Fig 5. Heat maps for Gene Ontology (GO) analysis of DEGs in terms of cellular components.** Gene ontology analysis was performed in Enrichr. The colors represent *p* values computed from Fisher's exact tests. Highlighted cellular compartments were significant changed in ZIKV infection.

related genes tended to increase, and DNA repair systems and cell cycle-related genes were downregulated resulting in cell death. The results supported that ZIKV disrupted the cell cycle of hNPCs by halting DNA replication during S phase and inducing DNA damage, while this does not occur for dengue virus [29]. Interestingly, DNA repair-related genes such as *TOP2B* (hPNs: −0.32, hNCCs: −0.46, mDCs: 1.80 $\log_2$ [fold-change]), *RPA2* (hPNs: −0.62, hNCCs: −0.44, mDCs: 1.57), and *NBN* (hPNs: −0.39, hNCCs: −0.46, mDCs: 1.78) in mDCs were activated as the defense mechanism against DNA damage, but not in neuronal cells. Dysregulation of pathways suggested that ZIKV has more serious effects on cell death in neuronal cells than in mDCs.

A recent study of ZIKV suggested that myeloid dendritic cells as well as neuronal cells play a very important role in pathology in ZIKV infection [11]. According to their results [11], mDCs appeared to be highly vulnerable to ZIKV infection and best-equipped for cell-intrinsic antiviral immune defense. Additionally, ZIKV-infected mDCs and non-infected mDCs from mixed populations exposed to ZIKV have different gene expression profiles.

ZIKV-infected mDCs may be a cellular component in the suppressed immune system by ZIKV infection, which may facilitate ZIKV infection in the host. Specifically, downregulated antiviral interferon-stimulated genes and innate immune sensors actively restrict interferon-dependent immune responses [30]. Additionally, envelope proteins and non-structural proteins (primarily NS1 and NS5) of ZIKV manipulate the expression of host cells to support viral immune escape by regulating the interferon pathway [31]. Based on comparative network analysis, the most distinguishable differences between the neuronal cells and mDCs were observed for the immune response including interferon signaling and prolactin signaling. Prolactin, which plays key roles in modulating the stress response, is secreted from the anterior pituitary gland and is synthesized in many extra pituitary tissues such as immune cells [32,33]. In the physiology of the immune system, prolactin signaling is known to act by stimulating the secretion of various cytokines and expression of cytokine receptors, and also as a growth and survival factor [34]. Prolactin signaling consisting of PI3K signaling and JAK/Stat signaling was downregulated only in mDCs as evidenced by downregulated *IRS2* (hPNs: 1.31, hNCCs: 1.16, mDCs: −0.35 log$_2$[fold-change]), *PIK3C3* (hPNs: 0.09, hNCCs: 0.45, mDCs: −1.20), *JAK3* (hPNs: −0.11, hNCCs: 0.40, mDCs: −1.95), *IRF1* (hPNs: 2.56, hNCCs: 1.77, mDCs: −1.41), and *STAT3* (hPNs: 0.41, hNCCs: 0.43, mDCs: −1.72). Also, NS5 protein of ZIKV is known to lead to immune suppression in host cells via proteasomal degradation of *STAT2*, which is a key factor in Type I interferon signaling [36]. From our analysis, *STAT3* may be a candidate for immune suppression by ZIKV infection, as several viruses reduced cellular *STAT3* by promoting proteasomal degradation [35], and NS1 protein of dengue-2 virus directly interacted with human *STAT3* protein [36]. Thus, downregulation of prolactin signaling as well as interferon signaling may be critical immune evasion strategies adapted by ZIKV.

Our results showed that in ZIKV-infected mDCs, the expression of phagocytic vesicle membrane (GO: 0030670) and clathrin-coated vehicle membrane (GO: 0030665 and GO: 0030118)-related genes was inhibited compared to in normal mDCs, suggesting that phagocytosis was impaired (Figs 2B and 5). Phagocytosis is a fundamental process for the clearance of foreign pathogens including viruses [37]. In mDCs, phagocytosis is performed very efficiently more than in any other cell type [38]. However, phagocytosis-involved pathways were inhibited in mDCs by ZIKV infection compared to that in neuronal cells. Because antigen-phagocytosis of mDCs is a critical process regulating a broad range of immune responses including T cell immune responses, down-regulated expression of genes associated with phagocytosis may affect a general defense mechanism against ZIKV in host immune system. Furthermore, genes related to antigen presentation including *HLA-B* and *HLA-C* were also suppressed (Fig 4A). Additionally, genes promoting immunogenicity such as *TLR1* (hPNs: 0.85, hNCCs: 1.31, mDCs: −1.16), *IRF1* (hPNs: 2.56, hNCCs: 1.77, mDCs: −1.41), and *RELA* (hPNs: 0.37, hNCCs: 0.26, mDCs: −0.09) may be downregulated in mDCs (Fig 4A) and the expression levels of neuroinflammation-related genes were generally decreased (Fig 4B). Taken together, the weakened functions of inducing antiviral immune responses through phagocytosis and antigen presentation for eliminating ZIKV may be an important mechanism of immune suppression against ZIKV.

Although there were limitations to this study based on the functional analysis of RNA-Seq data, we found that the activation of inflammation and the inhibition of DNA repair and cell-cycle pathways were potential molecular mechanisms leading to neuronal cell death by ZIKV infection. Additionally, mDCs may play essential roles in virus growth through an immune escape mechanism in ZIKV infection. Additional studies are needed to confirm the results of this study, which will provide important insights into the mechanisms of infection of ZIKV and resulting in neuronal cell death and mDC dysfunction.

## Supporting information

**S1 Table. List of 96 filtered genes with p values of less than 0.01 in statistical *t*-tests.**
(XLSX)

**S2 Table. Details of all DEGs in hPNs, hNCCs, and mDCs.**
(XLSX)

**S3 Table. Top 10 significantly up- or downregulated pathways in hPNs, hNCCs, and DCs.**
(XLSX)

**S1 Fig. Network with significant pathways (p value <0.05) common in hPNs, hNCCs, and mDCs.** Protein-protein interactions were grouped by significantly expressed biological pathways in hPNs (A), hNCCs (B), and mDCs (C). The nodes represent biological pathways, and node size depends on the betweenness centrality score calculated by network analysis. Edges were generated based on interactions of DEGs contained in pathways. Red: upregulated pathways; blue: downregulated pathways. The networks were generated with v3.7.0 Cytoscape.
(TIF)

**S2 Fig. The canonical pathway of prolactin signaling from IPA.** Prolactin signaling consists of JAK/Stat signaling, PI3K signaling, and RAS cascade.
(TIF)

## Author Contributions

**Conceptualization:** Myung-gyun Kang, Daeui Park.

**Data curation:** Tamina Park, Chang Hoon Lee, Daeui Park.

**Investigation:** Tamina Park, Seung-hwa Baek, Daeui Park.

**Project administration:** Myung-gyun Kang, Chang Hoon Lee.

**Resources:** Myung-gyun Kang, Seung-hwa Baek.

**Supervision:** Daeui Park.

**Validation:** Seung-hwa Baek, Chang Hoon Lee, Daeui Park.

**Visualization:** Tamina Park.

**Writing – original draft:** Tamina Park.

**Writing – review & editing:** Tamina Park, Chang Hoon Lee, Daeui Park.

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
