## [Decision Letter · Decision Letter 0]

2 Jan 2020

PONE-D-19-31016

Zika virus infection differentially affects genome-wide transcription in neuronal cells and myeloid dendritic cells

PLOS ONE

Dear Dr. Park,

Your manuscript has been reviewed by two experts in the field and their comments follow, It was felt that experimental validation of a selected group of key genes should be performed.

After careful consideration, we feel that your paper has merit but does not fully meet PLOS ONE’s publication criteria as it currently stands. Therefore, we invite you to submit a revised version of the manuscript that addresses the points raised during the review process.

We would appreciate receiving your revised manuscript by Feb 16 2020 11:59PM. To enhance the reproducibility of your results, we recommend that if applicable you deposit your laboratory protocols in protocols.io, where a protocol can be assigned its own identifier (DOI) such that it can be cited independently in the future. For instructions see: http://journals.plos.org/plosone/s/submission-guidelines#loc-laboratory-protocols

We look forward to receiving your revised manuscript.

Kind regards,

Dong-Yan Jin

Academic Editor

PLOS ONE

Journal Requirements:

3. Thank you for including the following funding information within the acknowledgements section of your manuscript; "This work was supported by the Korea Institute of Planning and Evaluation for Technology 449 in Food, Agriculture, Forestry and Fisheries (IPET) through Animal Disease Management 450 Technology Development Program, funded by Ministry of Agriculture, Food and Rural 451 Affairs (MAFRA) (116097033SB010) and National Research Council of Science & 452 Technology grant by the Korea government (CRC-16-01-KRICT)."

Additional Editor Comments:

All other comments raised by reviewers should be satisfactorily addressed.

Reviewers' comments:

Reviewer's Responses to Questions

**Comments to the Author**

1. Is the manuscript technically sound, and do the data support the conclusions?

Reviewer #1: Partly

Reviewer #2: Yes

2. Has the statistical analysis been performed appropriately and rigorously? 

Reviewer #1: Yes

Reviewer #2: Yes

3. Have the authors made all data underlying the findings in their manuscript fully available?

Reviewer #1: Yes

Reviewer #2: Yes

4. Is the manuscript presented in an intelligible fashion and written in standard English?

Reviewer #1: Yes

Reviewer #2: Yes

5. Review Comments to the Author

Reviewer #1: The manuscript entitled “Zika virus infection differentially affects genome-wide transcription in neuronal cells and myeloid dendritic cells” by Park et. al. analyzed RNA sequencing data from ZIKV-infected neuronal cells and myeloid dendritic cells by comparative network analyses using protein-protein interaction information, and revealed that some major genes related to DNA repair systems, prolactin signaling as well as the interferon signaling, neuroinflammation, and cell cycle pathways showed differential changes in the peripheral neurons, neural crest cells, and myeloid dendritic cells. These findings further enriched current knowledge about the pathogenesis of ZIKV infection, but some serious concerns still need to be addressed before publication.

1. Some gene set and pathways in this manuscript were earlier published, so the authors should point out and focus on the novel genes and pathways they found using comparative network analysis.

2. All results were acquired from network analysis, it's better to choose some top hit new genes and further validate them by experiments.

Reviewer #2: The manuscript by Park et. al. entitled “Zika virus infection differentially affects genome-wide transcription in neuronal cells and myeloid dendritic cells” describes how ZIKA virus infection affects the transcriptome of human pluripotent stem cells derived peripheral neurons, neural crest cells and myeloid dendritic cells by analyzing published RNA seq data. In this study authors did comparative network analysis by protein-protein interaction predication. Authors found that major gene set are affected by ZIKV infection is DNA repair systems and prolactin signaling as well as the interferon signaling, neuroinflammation, and cell cycle pathways. By reanalysis of data from previous submission, authors find some interesting immune and mitochondrial dysfunction genes in myeloid dendritic cells. Overall the manuscript is well written and authors tried to find out the novel pathways such as prolactin signaling and mechanisms of Zika virus infection in these cell types. But authors should need to take care of following points:

1. The figures 3, 4 and 5 in the manuscript are completely unable to understand and read properly to reach any conclusion.

2. In fig-1A, its looks like human pluripotent stem cells infected by ZIKV and then differentiated into neuronal cell types. But it should be like hPSC differentiated hPN and hNCC infected by ZIKV.

3. Authors should need to discuss in details about the genes involved in ZIKA infection mediated mitochondria dysfunction, oxidative phosphorylation and prolactin signaling genes (IRS2, PIK3C3, JAK3, STAT3, and IRF1) as mentioned in results sections.

6. PLOS authors have the option to publish the peer review history of their article (what does this mean?). If published, this will include your full peer review and any attached files.

Reviewer #1: No

Reviewer #2: Yes: Shashi Kant Tiwari

---

## [Author Response · Author response to Decision Letter 0]

27 Feb 2020

Dear Prof. Dong-Yan Jin.

Here is the second revision about "PONE-D-19-31016

Zika virus infection differentially affects genome-wide transcription in neuronal cells and myeloid dendritic cells".

Our paper is valuable to understand the interaction of host cells and viruses such as ZIKV. In the paper, we focused on the significant difference gene sets when ZIKV enter into neuronal cells and immune cell using comparative network analysis. The methodology and systemic approaches will be helpful to prevent emerging viruses such as SARS-CoV-2. In addition, RNAseq has a high correlation experimental qPCR with over 85%. We believe the result of ZIKV infection between neuronal cells and the immune cell can give new clues for developing ZIKV drugs and vaccines. 

Thank you for your services.

Sincerely yours,

Daeui Park

PONE-D-19-31016

Zika virus infection differentially affects genome-wide transcription in neuronal cells and myeloid dendritic cells

PLOS ONE

Reviewers' comments:

Reviewer's Responses to Questions

Comments to the Author

 1. Is the manuscript technically sound, and do the data support the conclusions?

 Reviewer #1: Partly

 Reviewer #2: Yes

 2. Has the statistical analysis been performed appropriately and rigorously? 

 Reviewer #1: Yes

 Reviewer #2: Yes

3. Have the authors made all data underlying the findings in their manuscript fully available?

 Reviewer #1: Yes

 Reviewer #2: Yes

 4. Is the manuscript presented in an intelligible fashion and written in standard English?

 Reviewer #1: Yes

 Reviewer #2: Yes

5. Review Comments to the Author

 Reviewer #1: The manuscript entitled “Zika virus infection differentially affects genome-wide transcription in neuronal cells and myeloid dendritic cells” by Park et. al. analyzed RNA sequencing data from ZIKV-infected neuronal cells and myeloid dendritic cells by comparative network analyses using protein-protein interaction information, and revealed that some major genes related to DNA repair systems, prolactin signaling as well as the interferon signaling, neuroinflammation, and cell cycle pathways showed differential changes in the peripheral neurons, neural crest cells, and myeloid dendritic cells. These findings further enriched current knowledge about the pathogenesis of ZIKV infection, but some serious concerns still need to be addressed before publication.

 1. Some gene set and pathways in this manuscript were earlier published, so the authors should point out and focus on the novel genes and pathways they found using comparative network analysis.

Our paper represented novel pathways as well as already published pathways in ZIKV infection. Major pathways are related to neuroinflammation, interferon signaling, and cell cycle pathways. In addition, we found out novel pathways and focused genes by ZIKV infection. 

Especially, prolactin signaling pathway including JAK/STAT, PI3K, RAS pathways were downregulated myeloid dendritic cell by ZIKV infection. The pathways were revealed by the comparison analysis between neuronal cells and mDCs, because the gene showed the highly different gene expression pattern. For example, IRS2 (hPNs: 1.31, hNCCs: 1.16, mDCs: −0.35 log2[fold-change]), PIK3C3 (hPNs: 0.09, hNCCs: 0.45, mDCs: −1.20), PDPK1 (hPNs: 0.43, hNCCs: 0.16, mDCs: −2.08), STAT3 (hPNs: 0.41, hNCCs: 0.43, mDCs: −1.72), and IRF1 (hPNs: 2.56, hNCCs: 1.77, mDCs: −1.41), etc. The more genes were also described on the result in manuscript.

Interestingly, DNA repair system including the NER pathway were only upregulated in mDC with TOP2B, USP7, CETN2, RPA2, and COPS2. Also, DNA Double-Strand Break Repair was upregulated in mDC, but not in neuronal cells. 

Additionally, oxidative phosphorylation and mitochondrial dysfunction share many genes which were significantly upregulated in mDCs, but not in neurons. Details of the genes are as follows. Oxidative phosphorylation, known as cellular respiration, occurs in the mitochondria, where enzymes catalyze the generation of ATP and the transfer of electrons to molecular oxygen. The upregulation of oxidative phosphorylation can lead to escape of the immune defense in mDCs from ZIKV infection.

 2. All results were acquired from network analysis, it's better to choose some top hit new genes and further validate them by experiments.

Network analysis can give us the benefit to understand the mechanism about interaction between viruses and host cells. Therefore, many approaches have applied network analysis as well as top hit method. Our paper was based on RNAseq data of same strain of ZIKV (PRVABC59) which isolated from Puerto Rico in 2015. Because the RNAseq methods have high gene expression correlations with qPCR data with over 85% [1], the genes could show consistent results between RNA-sequencing and qPCR.

Reviewer #2: The manuscript by Park et. al. entitled “Zika virus infection differentially affects genome-wide transcription in neuronal cells and myeloid dendritic cells” describes how ZIKA virus infection affects the transcriptome of human pluripotent stem cells derived peripheral neurons, neural crest cells and myeloid dendritic cells by analyzing published RNA seq data. In this study authors did comparative network analysis by protein-protein interaction predication. Authors found that major gene set are affected by ZIKV infection is DNA repair systems and prolactin signaling as well as the interferon signaling, neuroinflammation, and cell cycle pathways. By reanalysis of data from previous submission, authors find some interesting immune and mitochondrial dysfunction genes in myeloid dendritic cells. Overall the manuscript is well written and authors tried to find out the novel pathways such as prolactin signaling and mechanisms of Zika virus infection in these cell types. But authors should need to take care of following points:

 1. The figures 3, 4 and 5 in the manuscript are completely unable to understand and read properly to reach any conclusion.

As your comments, we changed figure 3 to clearly understand major gene set between neural cells and myeloid dendritic cells. Also, major gene were highlighted in figure 4 and 5.

 2. In fig-1A, its looks like human pluripotent stem cells infected by ZIKV and then differentiated into neuronal cell types. But it should be like hPSC differentiated hPN and hNCC infected by ZIKV.

Correct, we changed the figure 1A. Thank you for your points.

 3. Authors should need to discuss in details about the genes involved in ZIKA infection mediated mitochondria dysfunction, oxidative phosphorylation and prolactin signaling genes (IRS2, PIK3C3, JAK3, STAT3, and IRF1) as mentioned in results sections.

More detail the information of gene was added in results section. Already, interferon signaling is known to major gene set in ZIKV infection. However, prolactin signaling could be regarded as novel candidates by ZIKV infection. Prolactin signaling is related to stimulate the secretion of cytokines and expression of cytokine receptors in the immune system which is consisted of PI3K signaling and JAK/Stat signaling. Interestingly, the IRS2, PIK3C3, JAK3, STAT3, and IRF1 which belong to prolactin signaling was significantly downregulated only in mDCs, not neural cell. In addition, STAT3 may be a candidate for immune suppression by ZIKV infection, as several viruses reduced cellular STAT3 by promoting proteasomal degradation, and NS1 protein of dengue-2 virus directly interacted with human STAT3 protein. We discussed the points in result and discussion of the manuscript.

Reference

1. Everaert C et al. Benchmarking of RNA-sequencing analysis workflows using whole-transcriptome RT-qPCR expression data. Sci Rep. 2017 May 8;7(1):1559

---

## [Decision Letter · Decision Letter 1]

16 Mar 2020

Zika virus infection differentially affects genome-wide transcription in neuronal cells and myeloid dendritic cells

PONE-D-19-31016R1

Dear Dr. Park,

We are pleased to inform you that your manuscript has been judged scientifically suitable for publication and will be formally accepted for publication once it complies with all outstanding technical requirements.

With kind regards,

Dong-Yan Jin

Academic Editor

PLOS ONE

Additional Editor Comments (optional):

Reviewers' comments:

Reviewer's Responses to Questions

**Comments to the Author**

1. If the authors have adequately addressed your comments raised in a previous round of review and you feel that this manuscript is now acceptable for publication, you may indicate that here to bypass the “Comments to the Author” section, enter your conflict of interest statement in the “Confidential to Editor” section, and submit your "Accept" recommendation.

Reviewer #1: All comments have been addressed

Reviewer #2: All comments have been addressed

2. Is the manuscript technically sound, and do the data support the conclusions?

Reviewer #1: Yes

Reviewer #2: Yes

3. Has the statistical analysis been performed appropriately and rigorously? 

Reviewer #1: Yes

Reviewer #2: Yes

4. Have the authors made all data underlying the findings in their manuscript fully available?

Reviewer #1: Yes

Reviewer #2: Yes

5. Is the manuscript presented in an intelligible fashion and written in standard English?

Reviewer #1: Yes

Reviewer #2: Yes

6. Review Comments to the Author

Reviewer #1: (No Response)

Reviewer #2: Authors answered the all queries raised in previous round of revision. But still few contents in figures are not looks very clear.

7. PLOS authors have the option to publish the peer review history of their article (what does this mean?). If published, this will include your full peer review and any attached files.

Reviewer #1: No

Reviewer #2: No

---

## [Editor Report · Acceptance letter]

24 Mar 2020

PONE-D-19-31016R1 

Zika virus infection differentially affects genome-wide transcription in neuronal cells and myeloid dendritic cells 

Dear Dr. Park:

I am pleased to inform you that your manuscript has been deemed suitable for publication in PLOS ONE. Congratulations! Your manuscript is now with our production department. 

With kind regards,

on behalf of

Professor Dong-Yan Jin 

Academic Editor

PLOS ONE